# Therapeutic Efficacy of *Arnica* in Hamsters with Cutaneous Leishmaniasis Caused by *Leishmania braziliensis* and *L. tropica*

**DOI:** 10.3390/ph15070776

**Published:** 2022-06-22

**Authors:** Sara M. Robledo, Javier Murillo, Natalia Arbeláez, Andrés Montoya, Victoria Ospina, Franziska M. Jürgens, Iván D. Vélez, Thomas J. Schmidt

**Affiliations:** 1PECET-School of Medicine, University of Antioquia, Calle 70 # 52-21, Medellin 0500100, Colombia; javier.murillo@udea.edu.co (J.M.); natyac182@gmail.com (N.A.); edwin.montoya@udea.edu.co (A.M.); victoriaospina@hotmail.com (V.O.); ivan.velez@udea.edu.co (I.D.V.); 2University of Münster, Institute of Pharmaceutical Biology and Phytochemistry, PharmaCampus, Corrensstrasse 48, D-48149 Münster, Germany; franziska.juergens@uni-muenster.de

**Keywords:** *Arnica montana* L., Arnica tincture, natural products, sesquiterpene lactones, cutaneous leishmaniasis, antileishmanial drugs, *Leishmania braziliensis*, *Leishmania tropica*, *Mesocricetus auratus*

## Abstract

Leishmaniasis may occur in three different clinical forms, namely, visceral, mucocutaneous and cutaneous, which are caused by different species of trypanosomatid protozoans of the genus *Leishmania*. Pentavalent antimonials are the leading treatment for cutaneous leishmaniasis despite the hepatic, renal, and cardiac toxicity. In addition, the response of some *Leishmania* species to pentavalent antimonials is increasingly poorer, and therefore new and more potent therapeutic alternatives are needed. *Arnica montana* L., Asteraceae, is a traditional medicinal plant of Europe and preparations of its flowers are commonly used externally to treat disorders of the musculoskeletal system as well as superficial inflammatory conditions. Previous studies have shown that Arnica tincture (AT), an ethanolic extract prepared from the flowerheads of *Arnica montana* as well as isolated Arnica sesquiterpene lactones (STLs) have antileishmanial activity in vitro against *L. donovani* and *L. infantum*, as well as in vivo against *L. braziliensis*. In this work, we studied the in vitro cytotoxicity and antileishmanial activity of AT and STLs against both *L. braziliensis* and *L. tropica*. The in vivo therapeutic effect of AT was studied in hamsters with cutaneous Leishmaniasis (CL) caused by experimental infection with *L. braziliensis* and *L. tropica*. Furthermore, various semisolid Arnica preparations were also evaluated against *L. braziliensis*. The STLs and the AT possess a very high in vitro activity against both *Leishmania* species with median effective concentrations (EC_50_) ranging from 1.9 to 5.9 μg/mL. The AT was not cytotoxic for human tissue macrophages, skin fibroblasts, and hepatic cells. The therapeutic response of hamsters infected with *L. braziliensis* to the topical treatment with AT was 87.5% at a dose of 19.2 μg STL/2× day/60 d, 72.7% at doses of 19.2 μg STL/1× d/60 d and 67% at a dose of 38.4 μg STL/2× d/60 d. In turn, the therapeutic response in hamsters infected with *L. tropica* was 100% when treated at a dose of 19.2 μg STL/2× day/60 d and 71% at a dose of 38.4 μg STL/2× d/60 d. On the other hand, the effectiveness of treatment with glucantime administered intralesionally at a dose of 200 mg/every three days for 30 days was 62.5% for *L. braziliensis* and 37.5% for *L. tropica* infection. These results are promising and encourage the implementation of clinical trials with AT in CL patients as a first step to using AT as a drug against CL.

## 1. Introduction

Leishmaniasis is an infectious disease caused by several species of a kinetoplastid protozoan belonging to the genus *Leishmania*. The disease is classified into three main clinical forms: cutaneous leishmaniasis (CL) affecting the skin, mucosal leishmaniasis (ML) affecting mucosal tissue mainly of the oro-nasopharyngeal region, and visceral leishmaniasis (VL) affecting the liver, spleen and bone marrow. These clinical forms or manifestations of the disease depend on the *Leishmania* species causing the infection and, in addition, *Leishmania* species have a defined geographical localization. The most common clinical presentation of the disease is CL, which causes localized or disseminated skin lesions. Without treatment, these lesions may persist, resulting in severe tissue damage and socially stigmatizing disfigurement [1]; in some CL cases caused by species such as *L. tropica* or *L. major* in the Old World (encompassing Europe, Asia, and Africa), the lesions may resolve; however, this is not the case for lesions caused by *Leishmania* species of the *Viannia* subgenus present in the New World (Central and South America). Worldwide, an estimated 12 million people are infected with *Leishmania*, and between 0.7 and 1.2 million people are newly infected each year. Leishmaniasis is endemic in less-developed countries of tropical and subtropical regions, and is especially prevalent in communities of lower socioeconomic power. Despite the prevalence of CL worldwide and the psychological and social impact of the disease and the consequent economic cost, treatment options are scarce [2]. For the past 80 years, pentavalent antimonials (Pentostam™ and Glucantime™) have been the leading treatment for CL despite the fact that these drugs cause hepatic, renal, and cardiac toxicity, as well as other serious side effects [3]. In addition, the response of some *Leishmania* species mainly *L. braziliensis* (New world) and *L. tropica* (Old world) to pentavalent antimonials is increasingly poorer, and therefore new and more potent therapeutic alternatives are urgently needed.

Not only pentavalent antimonials, but also other anti-leishmanial drugs such as pentamidine isethionate and amphotericin B are administered via parenteral routes, often intravenously, which can create barriers to treatment in areas with insufficient health care providers and populations with limited access to health centers; as a consequence of incomplete treatment, some strains of *Leishmania* have begun to show resistance to pentavalent antimonial compounds [4,5]. Thus, the availability of topical treatment is an attractive option not only for overcoming the problems of toxicity, but also because its non-invasive nature and ease of use can potentially improve treatment accessibility and enhance patient acceptability and treatment adherence, thereby increasing cure rates in outpatient settings. 

*Arnica montana* L., Asteraceae, is a traditional medicinal plant, and its flowers are commonly used for the relief of bruises, sprains and localized muscular pain [6,7,8], mainly because of its well-documented anti-inflammatory activity [7,8]. Arnica tincture (AT) is an ethanolic tincture prepared according to the European Pharmacopoeia from the flowerheads of *A. montana* [9]. It is an approved and marketed herbal medicinal product in Europe, which is used traditionally for external treatment of disorders of the musculoskeletal system as well as superficial inflammatory conditions. AT is known to contain a complex mixture of sesquiterpene lactones (STLs) consisting mainly of helenalin and 11α,13-dihydrohelenalin esters (examples are shown in Figure 1), which are commonly considered mainly responsible for its well-documented therapeutic effects. Besides the STLs, Arnica flowers also contain various other types of secondary metabolites such as flavonoids, essential oil, carotenoids, phenolic acids, lignans and (non-toxic) pyrrolizidine alkaloids [7,8,10].

Arnica STLs have been demonstrated to possess strong activity against trypanosomatid parasites, including *L. donovani* and *L. infantum* (see literature cited in [11]). A previous study has also shown that AT has comparable or even better curative effects than the standard drug glucantime, in experimental CL in golden hamsters (*Mesocricetus auratus*) infected with *L. braziliensis* [11]. In the present study, we investigated the in vitro antileishmanial activity and cytotoxicity of AT and various of its isolated STL constituents against both *L. braziliensis* and *L. tropica,* causative agents of CL in the New and Old World, respectively. In particular, the in vivo efficacy of topical treatment with AT with experimental CL in golden hamsters caused by *L. braziliensis* and *L. tropica* was studied in detail. Various treatment schemes were investigated to find the optimal dosage and application frequency. Furthermore, several semisolid Arnica preparations were evaluated for in vivo efficacy against *L. braziliensis* infection in the golden hamster model and compared with AT. The residual parasite load in the skin of cured animals and the skin penetration of STLs was investigated. A possible contribution of a wound healing activity of AT to the overall observed effects was also studied.

## 2. Results and Discussion

### 2.1. In Vitro Cytotoxicity of AT and STLs against Human Macrophages, Skin Fibroblasts, and Liver Cells

The results of cytotoxicity assays with AT and various isolated STLs (see Figure 1) against human macrophages (U937), skin fibroblasts (Detroit 551), and liver cells (Hep G2) are reported in Table 1. AT showed only moderate cytotoxicity against the macrophages and no cytotoxicity against liver cells and skin fibroblasts. In turn, all isolated STLs were cytotoxic for macrophages and liver cells at relatively low concentrations (LC_50_ values of 10.3 µg/mL or lower for U937 macrophages and 52.2 µg/mL or lower in Hep G2 liver cells). In the case of skin fibroblasts, helenalin, helenalin tiglate, and 11α,13-dihydrohelenalin acetate were not cytotoxic (with LC_50_ values higher than 200 µg/mL) while helenalin acetate, helenalin isobutyrate, helenalin methacrylate and 11α, 13-dihydrohelenalin isovalerate were moderately cytotoxic with LC_50_ values of about 60–70 µg/mL. Thus, the toxicity of STLs for human skin fibroblasts was 10 to 40 times lower than for macrophages and 5 to 300 times lower than for liver cells. These results suggest that the cytotoxic potential of Arnica STLs and, in particular, of the AT, should be tolerable when applied topically, which is in agreement with traditional use of AT as a topical preparation. It is quite interesting to note that the tincture was found far less cytotoxic than would be the equivalent amount of isolated STLs. This result nicely shows that other constituents of the total ethanolic extract may modulate the STLs’ toxic effects which is also in agreement with previous observations with helenalin and flavonoids [12].

### 2.2. In Vitro Antileishmanial Activity

The antileishmanial activity of AT as well as various isolated Arnica STLs (Figure 1) was tested in intracellular amastigotes of *L. braziliensis* and *L. tropica* (Table 2). The EC_50_ is referred to as the concentration of AT or STLs that causes 50% mortality calculated by Probit. Both AT as well as single STLs possess a very high activity against both *Leishmania* species (Table 2). While the selectivity indices (SI) of the isolated STLs are rather low, the AT displays reasonable SI values > 15 in case of both parasites.

### 2.3. Response of Hamsters with CL Caused by L. braziliensis and L. tropica to Treatment with Arnica Tincture and other Formulations

To find an optimal therapy scheme, the effectiveness of AT in healing hamsters with CL caused by *L. braziliensis* was evaluated by topical application of AT in three groups of hamsters. Group **1** (*n* = 11 hamsters) received 40 µL of AT (corresponding to a dose of 19.2 μg total STL) once per day for 60 days, i.e., the total dose was 1.152 mg of STLs per animal. Group **2** (*n* = 8) received the same dose (40 µL AT = 19.2 μg total STL), but twice daily for 60 days (total dose = 2.304 mg STLs per hamster). Group **3** (*n* = 9) was treated with 80 µL AT (38.4 μg total STL) twice daily for 60 days (total dose = 4.608 mg of STLs per hamster). The treatment results are shown in Figure 2 and summarized in Table 3.

In group **1**, the treatment resulted in a complete cure with re-epithelialization of the skin in 8/11 hamsters after 90 days post treatment. In 2 of the 11 hamsters, 69.9% and 67.6% reduction in the lesion size compared to the lesion size before treatment was observed, while in one animal that failed at the end of the study there was a relapse of the ulcer after a preliminary cure (Figure 2A). In group **2**, 7/8 hamsters were cured at the end of the study and 1 hamster displayed an improvement by a 92% reduction of the lesion size (Figure 2B). In group **3**, 6/9 hamsters were cured at post treatment day 90 (PTD90), while 2/9 hamsters showed improvement of the lesions with a reduction of 27.8% and 30.6% in lesion size. One hamster of this group did not respond to the treatment (Figure 2C).

Since semisolid preparations such as gels, creams or ointments might offer better patient compliance in comparison to the liquid tincture, four such preparations of Arnica were included in the study. Two commercially available creams, namely, DOC Arnika 21,5% Creme, containing 21.5% AT as active ingredient (DAC; treatment group **6**) and Arnica Salbe S, Kneipp (ASK, group **7**), containing 10% of an oily extract from Arnica flowers (contents according to manufacturers’ specifications; the absolute STL contents were determined in this study, see Materials and Methods) were considered. Besides these, formulations with increased STL content were included as well, namely, an STL-enriched Arnica cream (EAC; group **8**) and an STL-enriched Arnica Gel (EAG; group **9**). The latter two products were prepared in the Münster laboratory. They were enriched to 4- fold and 4.8-fold STL content, respectively, in comparison with the commercial products. The effectiveness of these four formulations was evaluated in hamsters infected with *L. braziliensis*. To this end, each of the mentioned preparations was administered topically at a dose of 2× 40 mg per day, for 60 days. This amounted to daily STL doses of 8.0 µg (DAC and ASK), 32.0 µg (EAC) or 38.4 µg (EAG), i.e., total STL doses of 480 mg, 1.920 mg and 2.304 mg, respectively.

In group **6,** treated with DAC, 4/11 hamsters cured, 2/11 displayed an improvement of their lesion with a reduction in the size between 15% and 30% but in 5/11 cases, the treatment failed after an initial cure observed at the end of treatment (TD60) (Figure 2D). In group **7**, 6/11 hamsters treated with ASK were cured at the end of the study (PTD90) and in 2/11 hamsters displayed a reduction of lesion size by 75% and 86%, while 3/11 hamsters showed treatment failure (Figure 2E). In group **8** treated with the STL-enriched Arnica cream EAC, 3/5 hamsters were cured at the end of the study (PTD90) while one hamster failed and the fifth hamster showed an improvement by reducing the size of the lesion by 45% (Figure 2F). In group **9**, represented by six hamsters treated with EAG, four hamsters did cure and the two hamsters showed improvement with a reduction of lesion size in percentages of 20% and 40% (Figure 2G).

The effectiveness of the AT was also evaluated in hamsters with CL caused by *L. tropica* at doses of 40 µL AT, equal to 19.2 μg total STL, twice daily for 60 days (total dose = 2.304 mg STL/hamster), *n* = 8 (group **4**) and 80 µL (38.4 μg total STL), twice daily for 60 days (total dose = 4.608 mg STL/animal), *n* = 9 (group **5**). In the case of group **4**, all hamsters were cured at the end of the study (Figure 3A) while in group **5***,* 5/7 hamsters were cured at PTD90 but in 2/*7* animals, the treatment failed (Figure 3B).

In summary, the overall effectiveness of the AT in hamsters infected with *L. braziliensis* (groups **1**–**3**) was 72.2%, 87.55, and 66.7%, respectively. Thus, the treatment with 40 µL AT (19.2 μg total STL per lesion) twice per day over 60 days proved the best among the three chosen schemes. The efficacy of AT in treating CL caused by *L. tropica* was 100% and 71% in groups **4** and **5**. Group **4** was treated in the same way as group **2** above (40 µL AT = 19.2 μg total STL/2× d/60 d) whereas group **5** received the double dose, which proved somewhat less favorable (Table 3). The therapeutic response of the semi-solid formulations in hamsters infected with *L. braziliensis* was poorer than in the case of AT (Table 3). As might be expected, the higher dose of STLs in the EAC and EAG led to an improved therapeutic result in groups **8** and **9** (60 and 67% cure rate, respectively) over that obtained with the DAC cream containing AT as active constituent, but without STL enrichment (36% cure rate). However, the results of the enriched products were only slightly better than that obtained with the ASK cream (55% cure rate), which has the same STL content as DAC. This might have to do with differences in galenical formulation, but the exact reason for the much better performance of ASK over DAC is unclear at present.

On the other hand, the cure rates achieved with meglumine antimoniate (MA) were 62.5% and 37.5% for *L. braziliensis* (group **10**, Figure 2H) and *L. tropica* (group **11**, Figure 3C), respectively. It may hence be safely stated that all treatment schemes of AT performed better than the standard antimonial treatment (Table 3).

In our previous study, the therapeutic response to AT was evaluated only at one dose and over a shorter treatment period. Hamsters infected with *L. braziliensis* were treated with a single dose of 100 µL AT (approx. 48 µg total STL), once a day for 28 days. This led to a cure rate of 60%. Considering that the topical treatment, being a local treatment, requires more time to obtain similar results to those obtained with systemic treatment, a lower dose of 40 µL equaling 19.2 µg total STL was evaluated, but treatment time was doubled to 60 days (group **1**). Effectively, the cure rate increased to 72%. On the other hand, to determine if the cure rate could be further improved by increasing the frequency of administration and the dose, the same dose of 19.2 µg total STL as well as the double dose (38.4 µg) were evaluated with twice-daily applications for 60 days (groups **2** and **3**, respectively). It was found that by increasing the frequency to two applications of 19.2 µg total STL per day, the cure rate increased to 87.5%. However, a further increase by doubling the dose to 2× 38.4 mg total STL caused a decrease of the cure rate to 67%.

These results suggest that the optimal therapeutic dose of AT in the hamster model is 40 µL (19.2 μg of total STL) administered per lesion at a frequency of twice daily for 60 days. The data also suggest that the efficacy may increase when increasing frequency, but that there is a limit to the total applied dose, because doubling the dose led to a decline in curative effectiveness. At the same time, it is clear from the assessment of the efficacy of various semi-solid preparations, that the liquid AT is more effective than semi-solid preparations. This can be advantageous since AT is easier and cheaper to produce than any of the semisolid preparations.

When comparing the response of *L. braziliensis* and *L. tropica* to topical treatment with AT, it was found that *L. tropica* infection displayed an even better response since all hamsters in group **4** were cured at the end of the study. A difference was observed when AT was applied once daily, most hamsters at this time showed no response in terms of the ulcer size reduction or healing, while upon two daily applications, most hamsters have already reduced the size of the lesion and several have even healed when the treatment ended. On the contrary, the response of hamsters infected with *L. tropica* to intralesional treatment with meglumine antimoniate was very poor, with cure and improvement rates as low as 37.5% and 25%, respectively, but also with failure (including relapse or reactivation of the lesions) rates as high as 37.5%. This poor response to meglumine antimoniate has also been reported in patients with CL caused by *L. tropica* [13]. It thus appears that AT might represent a particularly attractive treatment option for these patients.

None of the treatments with Arnica or meglumine antimoniate significantly affected the weight of the hamsters during the study (Figure 2I and Figure 3D). Furthermore, no alterations in serum enzyme AST, BUN and creatinine levels were found in any of the hamsters in the different treatment groups (Table 4) nor macroscopic alterations were observed in any of the organs or tissues after treatment.

One advantage of this experimental model of CL in golden hamster dorsal skin over other animal models such as, e.g., infections on the ears or footpads of mice, is that hamsters respond to *Leishmania* infection in a similar manner as humans, developing lesions of different sizes. Likewise, the response to treatment is similar to that observed in human patients, with some individuals healing or showing improvement of their lesions, but also responding faster or slower, or even responding negatively to treatment, i.e., showing failure as an outcome [14].

The effectiveness observed here for AT makes it a promising alternative for the local treatment of CL. Due to the risk of toxicity of systemic therapy, local treatment is recommended as an option to treat CL caused by all Old World species, but also to treat cases of uncomplicated CL caused by New World species. The risk of patients with New World CL developing ML even many years after the initial episode, which has been the main argument against local treatment, has been reevaluated [15].

### 2.4. Skin Penetration of STLs from AT and Semisolid Preparations

To determine the skin penetration of STLs from AT, the STL-enriched cream (EAC) and the gel (EAG), an ex vivo study with Franz diffusion cells and skin samples of golden hamsters was performed. After 24 h, the experiment was terminated and the skin was washed with PBS to remove the not absorbed STLs from the skin surface. In these skin wash solutions, only 2.7 ± 1.1% of the applied STL amount was quantified in the case of Arnica tincture. In contrast, 16.4 ± 0.1% and 12.3 ± 1.4% were quantified in the skin wash solutions after application of the STL-enriched ointment or gel, respectively. It can be concluded that the rest of the applied STLs penetrated into the skin. Consequently, 97.3%, 83.6% and 87.7% of the STL amount penetrated into the skin after the application of the AT, EAC and EAG, respectively. These results might explain why the efficacy of the Arnica tincture treatment was higher compared to the treatment with the semisolid preparations.

### 2.5. Residual Parasite Load in Cured Animals’ Skin

To investigate the presence of parasites in the skin of healed animals in response to treatment with AT or semi-solid Arnica formulations, the parasite load was determined in the cured animals of groups **2**, **3**, **8** and **9**
*(L. braziliensis*) and groups **4** and **5** (*L. tropica*), as well as in hamsters cured from *L. braziliensis* and *L. tropica* by treatment with meglumine antimoniate, by real time quantitative polymerase chain reaction (RT-qPCR). The average number of remaining parasites after treatment with AT was extremely low with 0.83 ± 1.32 and 3.20 ± 3.83 parasites per gram of tissue, in groups **2** (*n* = 7) and **3** (*n* = 6), respectively. For groups **4** (*n* = 6) and **5** (*n* = 5), the average numbers of parasites per gram of tissue were 0.42 ± 1.94 and 4.31 ± 2.16, respectively. On the other hand, in groups **8** and **9**, treated with the STL-enriched cream and gel, respectively, the remaining parasite load was somewhat higher with 15 ± 17 (*n* = 3) and 5.5 ± 6.0 (*n* = 4) parasites per gram tissue. This result appears to agree with the somewhat lower healing efficacy of these preparations in comparison with AT. In turn, the average number of parasites in healed skin of hamsters infected with *L. braziliensis* and *L. tropica* and treated with meglumine antimoniate was 5.42 ± 3.94 and 6.52 ± 7.82 parasites per gram of tissue. This finding indicates that despite clinical healing evidenced by scar formation, a few parasites may persist at the site of injury after the treatment with AT as well as that observed for pentavalent antimony. These results also confirm that clinical cure is not necessarily associated with sterile cure as reported elsewhere [16]. Due to the implications of the persistence of parasites for the clinical evolution and relapses, a piece of the cicatrized skin from every cured animal in groups **2**–**5**, **8** and **9** as well as both groups treated with meglumine antimoniate was cultivated in NNN medium to investigate the viability of parasites if present. None of the samples corresponding to groups **2**, **3**, **4**, **8** and **9** showed parasite growth. Only in one skin sample from group **5**, but in all the skins of the meglumine antimoniate treated hamsters, were promastigotes observed to grow after 4 weeks in the culture medium. The results show that although in some skins their RNA can still be detected after Arnica treatment, the parasites were not viable anymore and could not grow in the culture medium.

### 2.6. Expression of Growth Factors in the Skin of Cured Animals

The fibroblast growth factor (FGF), platelet-derived growth factor (PDGF), transforming growth factor-beta 1 (TGFβ1) and alpha (TGFα), epidermal growth factor (EGF), connective tissue growth factor (CTGF), and vascular endothelial growth factor (VEGF) are polypeptides that stimulate tissue repair [17,18]. A previous study demonstrated that in golden hamsters with CL caused by *L. braziliensis*, the relative expression levels of EGF increased in skin biopsies of cured hamsters [19]. In this work, the relative expression levels of EGF, FGF, and TGFβ1 were observed in skin biopsies of cured hamsters after treatment with AT (Figure 4). EGF levels were 3.72 times higher than TGFβ1 and 2.44 times higher than FGF levels. In turn, levels of expression of TGFβ1 were 1.52 times higher than FGF levels. These results are in agreement with those previously reported where increased EGF expression levels were observed in hamsters that responded positively to antileishmanial treatment [19].

Wound healing is the summation of a complex series of processes beginning with coagulation and ending in remodeling. This process is associated with cell migration and proliferation, extracellular matrix remodeling, angiogenesis, and re-epithelialization, which involve cells, growth factors and other immunological and biochemical factors. Both EGF and FGF are secreted by platelets, macrophages, and fibroblasts [20] while FGF is produced by keratinocytes, fibroblasts, endothelial cells, smooth muscle cells, chondrocytes, and mast cells [21] and induces migration of fibroblast into the wound during inflammation. Both EGF and FGF act in a paracrine fashion on keratinocytes and stimulate epithelial cell migration and proliferation of keratinocytes over the provisional extracellular matrix [22]. Once wound closure (100% epithelialization) is achieved, keratinocytes undergo stratification and differentiation to restore the barrier [20] and increase in the acute wound, and play a role in granulation, tissue formation, re-epithelialization, and tissue remodeling [21]. Despite the importance of growth factors in wound repair, to date, their role in the healing process in CL has not been studied.

### 2.7. Possible Influence of AT on Wound Healing

In order to study the potential contribution of an influence of AT on wound healing to the overall observed curative effects, scratch test experiments were carried out. Exposure of human fibroblasts (Detroit 551 cells) to 200 µL AT containing 96 μg total STLs for 24 h reduced the gap area by 22%, 37.3% and 65% after 8 h, 16 h, and 24 h of incubation while untreated cells, cultured in a medium with 10% fetal bovine serum (FBS) reduced the gap area by 12.7%, 13.4%, and 42% after the same time intervals, respectively (Figure 5). The effect induced by the AT on gap closure, i.e., cell migration, was statistically significant at all times evaluated (*p* < 0.001). These results suggest that AT significantly promotes skin healing under the chosen in vitro conditions.

## 3. Materials and Methods

### 3.1. Arnica Tincture and STLs

The Arnica tincture (AT) under study was a commercial product marketed in Germany under the name Arnikatinktur Hetterich (Manufacturer: Teofarma SRL, Valle Salimbene, Italy). All experiments reported here were performed with batch number 144901). The STL content was determined by an UHPLC-(+)ESI QqTOF MS method (see Appendix A), which was developed and fully validated for this study. The analytical details of this method have been published very recently [23]. The STLs were isolated based on the description in [24] and with a purity of >95% (determined by ^1^H-NMR spectroscopy, see Appendix A).

The total content of STLs in the AT under study was determined as 480.56 ± 34.87 μg/mL (*n* = 3).

### 3.2. Arnica Formulations

Two commercial semisolid Arnica preparations were purchased and included in the study: Arnica Salbe S (Kneipp) (Manufacturer: Kneipp GmbH, Würzburg, Germany; batch number 1804434; ASK) and DOC Arnica 21,5% Creme (Manufacturer: Hermes Arzneimittel GmbH, Pullach, Germany; batch number 8080328; DAC). Their STL contents were analyzed as described below. ASK was found to contain 0.10 ± 0.02 mg/g, whereas DAC contained 0.10 ± 0.01 mg/g total STLs.

The STL-enriched Arnica cream (EAC) was prepared by adding 8.9 g of 10× concentrated AT to 90.5 g of the commercial cream (ASK, see above). The 10× concentrated Arnica tincture was achieved by rotary evaporation under reduced pressure to 1/10 of its original volume. The final total STL concentration was determined at 0.40 ± 0.02 mg/g.

The Arnica gel with enriched STL content (EAG) was produced with 70.1 g of 1.33× concentrated Arnica tincture (obtained by rotary evaporation of Arnica tincture to ¾ of its original volume under reduced pressure), 20.0 g methylcellulose as a gelling agent and 10.0 g propylene glycol as a humectant. The final total STL concentration was determined as 0.48 ± 0.01 mg/g.

Analysis of the STL content in semisolid preparations: The STL amounts in ASK and DAC were analyzed with the UHPLC-(+)ESI QqTOF MS method (see above) after the following sample preparation in triplicates based on [25]. First, the internal standard α-santonin (0.1 mg) was added to 5 g of the formulations, and extraction was performed five times with 15 mL acetone each. Next, combined acetone extracts were evaporated to dryness and resolved in 10 mL of petroleum ether (40–60 °C). Then, liquid–liquid extraction with 15 mL of 65% ethanol was performed four times, combined ethanol phases were evaporated to dryness and samples were resolved in acetone (2.5 mL) and purified with amino propyl solid phase extraction cartridges. The cartridges were conditioned with 2 mL of acetone prior to sample application and acetone/ethyl acetate (1/1, *v*/*v*) was used as eluent (2 mL). Eluates were dried and resolved in 1 mL methanol for UHPLC-(+)ESI QqTOF MS analysis.

To verify the STL amounts in the generic in-house-produced formulations EAC and EAG, the gel as well as the STL-enriched cream were analyzed with two replicates for each formulation. The sample of EAC was prepared for UHPLC-(+)ESI QqTOF MS analysis as described above for the commercial product ASK. For the analysis of the gel EAG, a different sample preparation was described by Wagner and Merfort [25]. After addition of the internal standard α-santonin (0.1 mg) to the gel (5.0 g), samples were evaporated under reduced pressure, dissolved in water (10 mL) and extracted with ethyl acetate (3× 20 mL). Combined ethyl acetate phases were evaporated under reduced pressure, leached with dichloromethane (2× 4 mL) and filtered. The filtrate was evaporated under reduced pressure and residues were dissolved in 4 mL methanol (50 Vol-%). After addition of 1.4 g neutral aluminum oxide and shaking, the samples were centrifuged and filtrated. To fit the UHPLC-(+)ESI QqTOF MS calibration curve range, 44 μL of the filtrate were evaporated under reduced pressure and dissolved in 1 mL methanol.

### 3.3. Human Cells and Culture

Cytotoxic activity (see Section 3.6) was evaluated in human promonocytes (U937, CRL-1593.2), human liver cells (Hep G2, HB-8065), and human skin fibroblasts (Detroit 551, CCL-110) (American Type Cell Collection-ATCC, Manassas, VA, USA). Cells were maintained cultured under standard conditions in RPMI-1640 (U937) or EMEM (Hep G2, Detroit 551) medium, enriched with 10% inactivated fetal bovine serum (FBS) and 1% antibiotics (100 U/mL penicillin and 0.1 mg/mL streptomycin), referred to as “complete” medium. All cells were incubated at 37 °C in a 5% CO_2_ atmosphere.

The U937 cells have the morphology of monocytes with a round appearance and grow as suspension cells. Hep G2 cells are epithelial cells of varying size and shape, but predominantly polygonal in shape; Detroit 551 cells are fibroblast-like cells, variable in shape and size; they may be spindle-shaped or stellate with cytoplasmic extensions that may be relatively short and broad, or long, thin and highly branched. Both Hep G2 and Detroit 551 cells grow as adherent cells forming a monolayer. Once the cell cultures are exposed to cytotoxic concentrations, the appearance of the U937 cells becomes irregular, with stellate and rough edges, and the Hep G2 and Detroit 551 cells detach from the monolayer and lose their extensions.

### 3.4. Parasites and Culture

*L. braziliensis* strain (MHOM/CO/88/UA301) and *L. tropica* (MHOM/SU/74/K27) both transfected with the green fluorescent protein (GFP) gene were used [26]. Parasites were maintained in the promastigote stage cultured in Novy-MacNeil-Nicolle (NNN) and PBS biphasic medium with glucose, pH 6.9 and incubation at 26 °C [27]. Promastigotes were used in U937 cell and hamster infection assays.

### 3.5. Animals and Housing

Therapeutic response studies were performed in golden hamsters (*Mesocricetus auratus*) purchased from Charles River Laboratories International, Inc. (Wilmington, MA, USA) and bred in the specific pathogen-free (spf) biotherium of the University of Antioch. The animals were maintained in a controlled environment of relative humidity and photoperiod. The animals were fed ad libitum with a standard diet (LabDiet 5010^®^) and drinking water, sterilized in an autoclave. 

The animal study protocol was approved by the Institutional Ethics Committee of Universidad de Antioquia (Act No. 153 of 6 October 2020).

The ARRIVE (Animal Research: Reporting of In Vivo Experiments) guidelines were followed to report research using animals (https://www.nc3rs.org.uk); accessed 21 June 2022).

### 3.6. In Vitro Cytotoxicity Assay

U937, Detroit 551, and Hep G2 cells were dispensed into each well of a 96-well plate at a density of 10,000 cells. After 24 h of incubation, six 1:2 dilutions of AT were added. Dilutions were prepared with the corresponding culture medium (EMEM for Hep G2 and Detroit 551 or RPMI-1640 for U937 cells) starting from an initial concentration of 100%, i.e., 50–25–12.5–6.25–3.125–1.56%, *v*/*v*. The STLs and MA were assessed at concentrations of 50–25–12.5–6.25–3.125–1.56 μg/mL. Cytotoxicity was assessed at 72 h using the 3-(4,5-dimethylthiazol-2yl)-2,5-diphenyl tetrazolium bromide (MTT) assay (Sigma-Aldrich, St. Louis MO, USA) and following the methodology described elsewhere [28]. Cells cultured under the same conditions in the absence of AT or STLs were used as viability (non-toxicity) controls while cells exposed to doxorubicin were used as toxicity control. Cytotoxicity was determined according to the percentage mortality for each concentration of the AT or STLs, according to the optical densities (O.D.) obtained in each experimental condition and in comparison with the O.D. obtained with cells not exposed to the extracts, using Equation (1):% Mortality = 1 − [O.D. cells exposed to the compound ÷ O.D. unexposed cells] × 100(1)

The relationship between the concentration of the AT (based on the content of total STLs) and the isolated STLs (*x* axis) with the percentage of inhibition (*y* axis) was established, and the values of the LC_50_ were calculated using the statistical program Prism 8 (GraphPad Software Inc., San Diego, CA, USA). Two assays were performed, with three replicates for each concentration tested.

### 3.7. Infection of Human Macrophages (U937) with L. braziliensis and L. tropica and Exposure to Arnica Tincture and STLs

U937 cells (3 × 10^5^ cells/mL of complete RPMI-1640 medium and 0.1 μg/mL phorbol myristate acetate (PMA) (Sigma-Aldrich, St. Louis MO, USA) were infected with *L. braziliensis* or *L. tropica* promastigotes in the early stationary growth phase (day 5 of culture) at a concentration of 45 × 10^5^ parasites/mL of complete RPMI-1640 medium. After 3 h of incubation at 34 °C, 5% CO_2_, extracellular parasites were removed by washing with PBS. Fresh medium was added to each well, and the plates were again incubated at 34 °C, with 5% CO_2_. After 24 h, the medium was replaced with a fresh medium containing the corresponding concentration of the corresponding AT or STL dilution. To ensure cell viability during the assay, the maximum concentration evaluated was twice the LC_50_. After 72 h of incubation at 34 °C, 5% CO_2_ the cells were read in a flow cytometer (Cytomics FC 500MPL, Beckman Coulter, Brea, CA, USA) following the methodology described [26]. Non-treated infected cells were used as control of infection while infected cells exposed to amphotericin B (Sigma-Aldrich, St. Louis MO, USA) were used as an internal control for leishmanicidal activity. Two assays were performed, with three replicates for each concentration tested. The leishmanicidal activity was determined according to the number of parasites in infected cells obtained for each AT/STLs concentration according to the number of positive events for green fluorescence in a dot plot, as well as according to the mean fluorescence intensity (MFI) in a histogram (see Appendix A). The decrease in the number of parasites (inhibition of infection) due to the effect of the AT or STLs was calculated using Equation (2): % Inhibition of infection = 1 − [IFM infected and exposed cells ÷ IFM infected and unexposed cells] × 100(2)

The values of the median effective concentration (EC_50_) were calculated using the statistical program Prism 8 (GraphPad Software Inc., San Diego, CA, USA). Cytotoxic activity was correlated with leishmanicidal activity by calculating the Selectivity Index (SI) by dividing the value of LC_50_ by the value of EC_50_ (SI = LC_50_ ÷ EC_50_).

### 3.8. Intradermal Infection of Golden Hamsters with L. braziliensis and L. tropica and Therapeutic Schemes

One hundred hamsters were experimentally infected with *L. braziliensis* or *L. tropica* promastigotes (1 × 10^8^) in stationary phase growth in 100 μL of PBS by intradermal injection in the dorsum. Prior to topical treatment, disease progression occurred as expected and hamsters developed ulcers about seven or eight weeks after infection. Once ulcers developed, hamsters were randomly distributed into 11 groups (*n* = 6 to 11, each group). To follow the progress of infection before, during and after treatment, the lesion size was measured using a digital caliper and the progress was documented in photographs during treatment and 90 days after. 

Three groups of hamsters infected with *L. braziliensis* were treated with AT at a dose of 19.2 μg/1× d/60 d (group **1**), 19.2 μg/2× day/60 d (group **2**), and 38.4 μg/2× d/60 d (group **3**). Two further groups of hamsters infected with *L. tropica* were treated with AT at a dose of 19.2 μg/2× day/60 d (group **4**) and 38.4 μg/2× d/60 d (group **5**). In addition, two groups of hamsters infected with *L. braziliensis* were treated with DOC Arnica cream (DAC; group **6**) or Arnika Salbe S Kneipp (ASK; group **7**); both 40 mg (=4.0 µg total STL)/2× d/60 d. Two additional groups of hamsters infected with *L. braziliensis* were treated with STL enriched cream (EAC, group **8**) and with STL enriched gel (EAG; group **9**); both 40 mg (=16.0 and 19.2 µg total STL, respectively)/2× d/60 d.

Two groups of animals were infected with *L. braziliensis* and *L. tropica* and treated with meglumine antimoniate, intralesionally (200 μg/every 3 days/30 d; positive control groups). After the end of treatment, hamsters were followed-up for three months. Animal welfare was monitored daily during the study for changes in behavior, food and water consumption, and urine and fecal appearance, and the effectiveness of each treatment was determined by comparing lesion sizes before (TD0) and after treatments at the following times: end of treatment (TD60) and at post treatment days 30, 60 and 90 (PTD30, PTD60, and PTD90, respectively) counted from the last day of treatment [14]. The outcomes considered as a result of treatment at the end of the study were: cure (100% cure of the area and complete disappearance of the lesion); improvement (any percentage reduction in lesion area) and failure (any increase in lesion size compared to baseline size). The toxicity of the Arnica products applied during 60 days was determined according to changes in body weight obtained during and after treatment and serum enzymes (alanine aminotransferase (ALT), alkaline phosphatase (AP), blood urea nitrogen (BUN) and creatinine levels before treatment (TD0) and on day 8 after treatment (PTD8)).

### 3.9. Estimation of Parasite Load in Healed Skin Specimens by RT-qPCR

The amount of parasite was determined in skin biopsies from the cured hamsters by RT-PCR using the Vero 1-step RT-qPCR SYBR Green kit. For this, the RNA from the skin biopsies was extracted using Trizol^®^ (Invitrogen, Waltham, MA, USA) following the manufacturer’s instructions and then quantified in a Nanodrop 1000 (Thermo Scientific, Waltham, MA, USA). Then, 20 ng of RNA, 12.5 μL of master mix solution, and 100 ng of each primer Fw 5′- TGAGCGCATCGAGTACCT -3′ and Rv 5′- TCCCGCTTGCCATCCTC -3′, with a volume adjusted to 25 μL using nuclease-free water, were used to carry out the reaction with the Smart Cycler II (Cepheid, Sunnyvale, CA, USA): 50 °C 15 min, 95 °C 15 min and 40 cycles 95 °C 15 s, 60 °C 20 s and 72 °C 20 s, a final cycle 72 °C 300 s and a melting curve between 60 and 95 °C. Absolute quantification was performed using a standard logarithmic scale from 1 to 1 million parasites [29]. This determination was carried out with skin samples of all cured animals, i.e., from 31 hamsters of groups **2**, **3**, **4**, **5**, **8** and **9** as well as eight skin samples from hamsters infected with *L. braziliensis* and *L. tropica* and treated with meglumine antimoniate.

### 3.10. Expression of Growth Factors Associated with Wound Healing

The expression of growth factors associated with wound healing was quantified in skin biopsies from the cured hamsters by RT-PCR using the methodology described elsewhere [19]. The RNA from the skin samples was extracted using Trizol^®^ (Invitrogen) following the manufacturer’s instructions and then quantified in a Nanodrop 1000 (Thermo scientific). For the retrotranscription process, 100 ng of RNA were treated with 1 μL DNase I (Fermentas, VWR International, Radnor, PA, USA), 1 μL of buffer, and 8 μL of nuclease-free water; the mixture was incubated in a thermocycler PTC 100TM (MJ research Inc., St. Bruno, QC, Canada) for three cycles: 30 min at 37 °C, 5 min at 4 °C, and 10 min at 65 °C. Using the maximum first-strand cDNA synthesis kit, the RNA was transcribed into cDNA, following the manufacturer’s instructions: 4 µL master mix reaction, 2 µL RT enzyme mix, 2 µL RNA DNase I treated, and 12 µL water were mixed and incubated in the PTC 100TM (MJ research Inc., St. Bruno, QC, Canada) thermocycler for three cycles: 10 min at 25 °C, 15 min at 50 °C, and 5 min at 85 °C.

Growth factor expression: Specific primers for EGF, FGF, and TGFβ, as well as fluorescent fluoresceine amidite (FAM)-labeled hydrolysis probes, were designed. Actin was used as constitutive gene. The sequences for each forward (Fw) and reverse (Rv) primer and probe used were as follows: 

FGF, Fw, 5′- GTGTCAAGGCTGCTAGGTTT -3′, Rv, 5′- ACACATTGTATCCATCCTCAA -3′ and probe 5′- TCGCCTCACTTCGATCCCG -3′; EGF, Fw, 5′- CAGAACAAAGCCAGAAAATC -3′, Rv, 5′- CTGCAAGTACGTTCGTTTAACT -3′ and probe 5′- AGACTCGCGTTGCAAGGCG -3′; TGFβ1, Fw, 5′- AGCCTGGACACACAGTACAGTA -3′, Rv, 5′- CTTGCGACCCACGTAGTAC -3′ and probe 5′- AACACAACCCGGGTGCTTC -3′; γActin, Fw, 5′- ACAGAGAGAAGATGACGCA -3′, Rv, 5′- GCCTGAATGGCCACGTAC -3′ and probe 5′- TTGAAACCTTCAAATGACGCA -3′. 

The reaction of amplification was carried out with 1 μL of the cDNA and using the following protocol: 600 s at 95 °C, 40 cycles of 15 s at 95 °C, and 60 s at 60 °C in a Smart Cycler II (Cepheid, Sunnyvale, CA, USA). The efficiency of the amplification reaction was determined using the LinReg program and the expression levels were calculated using the ΔΔCT method, comparing the levels of expression on post treatment day 90 (PTD90) with those before treatment (TD0). This trial was conducted in skin samples from 11 hamsters treated with the dose of AT that yielded the best curative response (groups **2** and **4**).

### 3.11. In Vitro Wound Healing Assay (Scratch Test)

To identify the effect of AT on wound closure, the CytoSelect ™ Wound Healing Kit (Cell Biolabs Inc., San Diego, CA, USA) was used following the methodology described [30]. Briefly, Detroit 551 fibroblasts (2.5 × 10^5^/mL) in complete DMEM medium were seeded in each well containing the inserts to demarcate the gap in the cell monolayer. Three wells were left without insert to allow monolayer formation (100% confluence). The dishes were incubated for 24 h. After removing the inserts from the wells, the cells were treated with AT (200 µL corresponding to 96 μg of total STLs). The plates were again incubated for 8 h, 16 h and 24 h at 37 °C, 5% CO_2_. The effect of the AT on gap closure was determined according to fibroblast migration in the gap zone by microscope photographs at 8 h, 16 h and 24 h. The percentage of gap closure was calculated compared to the migration observed in the negative control well (untreated cells cultured in DMEM medium with 10% SFB). For quantification of the healing effect, the percentage of gap closure was calculated using Equation (3): Gap closure (%) = [(test compound (%) − untreated control (%))/(confluent area (%) − untreated control (%))] × 100(3)

### 3.12. Skin Penetration Experiments with Golden Hamster Skin in Franz Diffusion Cell

The skin penetration of STLs from AT, STL-enriched Arnica ointment and Arnica Gel formulations were tested using the skin of golden hamsters. The receptor fluid consisted of: 70% saline (12%)/30% Ethanol/0.6% Tween 20. The Franz cell was loaded with skin pieces of 3.14 cm^2^ and maintained at a constant flow rate and temperature of 32 ± 1 °C. The Franz cell was loaded with 10 mL of the receptor fluid, placed in contact with the skin, and allowed to acclimatize for 1 h in the Franz cell. Subsequently, 100 mg of ointment, 100 mg of gel, and 500 µL of AT (equivalent to 40, 48 and 240 μg total STL, respectively) were applied, and the assay was started. Samples (1 mL) from the receptor fluid were taken after 1 h, 3 h, 6 h, and 24 h, restoring the volume with 1 mL of receptor fluid at each sampling. At the end of the assay (24 h), the skin was washed with 10 mL of PBS. Then, the *stratum corneum* was removed by carefully pulling it off with 14 strips of adhesive tape; the remaining skin and the remaining liquid from the diffusion cell were also saved. Each assay was performed in duplicate.

Analysis of Franz cell samples was performed as described in [23]. First, the internal standard α-santonin (10 µg/mL in methanol) was added to each sample (10 µL to receptor fluids, skin on adhesive tape and remaining skin samples and 30 µL to skin wash solutions). Next, the adhesive tape strips were extracted four times with 5 mL methanol for 1 h each. Likewise, the remaining skin samples were extracted four times with methanol (5 mL) for 1 h each after they were cut into small pieces.

The receptor fluids and skin wash solutions were desalinated by solid phase extraction (SPE) with Oasis HLB cartridges (Waters Corporation, Milford, MA, USA). Therefore, the receptor fluid samples were diluted (1:2) with distilled water to reduce the organic portion for the subsequent SPE. Then, the SPE cartridges were conditioned and equilibrated with 0.5 mL of methanol and 0.5 mL of distilled water, respectively. Next, the samples (3 mL of diluted receptor fluid or 5 mL of skin wash solution) were loaded on the cartridges and washed twice with 0.4 mL (receptor fluids) or 0.8 mL (skin wash solutions) of distilled water. For the elution of STLs 0.2 mL of acetonitrile/methanol (9/1, *v*/*v*) was used.

For the LC-MS analysis, all SPE eluates (receptor fluid samples and skin wash solution samples) and methanolic extracts (adhesive tape with *stratum corneum* samples and the remaining skin samples) were dried and the residues were dissolved in 250 µL of water/acetonitrile (95/5, *v*/*v*). The LC-MS parameters are described in [24]. An injection volume of 20 µL was used. To evaluate the results, the extracted ion chromatograms (EICs) at *m*/*z* 245.1172 (helenalin derivatives) and *m*/*z* 247.1329 (11α,13-dihydrohelenalin derivatives) were used, as these are the most abundant *m*/*z* values of the STLs. Unfortunately, no quantification of STLs was possible in the skin samples and receptor fluids because of an interfering signal at *m*/*z* 245.1172.

### 3.13. Statistical Analysis

The differences between the parasitic loads according to the clinical results were determined by a one-way ANOVA and a Tukey’s test for multiple comparisons.

## 4. Conclusions

The results of this study provide unequivocal evidence that topical treatment of experimental CL in golden hamsters with Arnica preparations leads to better cure rates than that obtained by intralesional injection of the standard drug meglumine antimoniate. Comparison of various treatment schemes using Arnica tincture (AT) showed that treatment with 40 µL AT per lesion twice per day over 60 days yields optimal results with very high cure rates of 87.5 and 100% in *L. braziliensis* and *L. tropica* infections, respectively, which is much better than in the case of meglumine antimoniate (62.5 and 25% cure rate, respectively). The results with the liquid ethanolic tincture were also better than those obtained with several semisolid Arnica preparations tested. Arnica tincture thus appears to be a very interesting candidate for a new treatment of CL whose therapeutic efficacy will next have to be tested in studies with human patients. Such studies have already been initiated.

## Figures and Tables

**Figure 1 pharmaceuticals-15-00776-f001:**
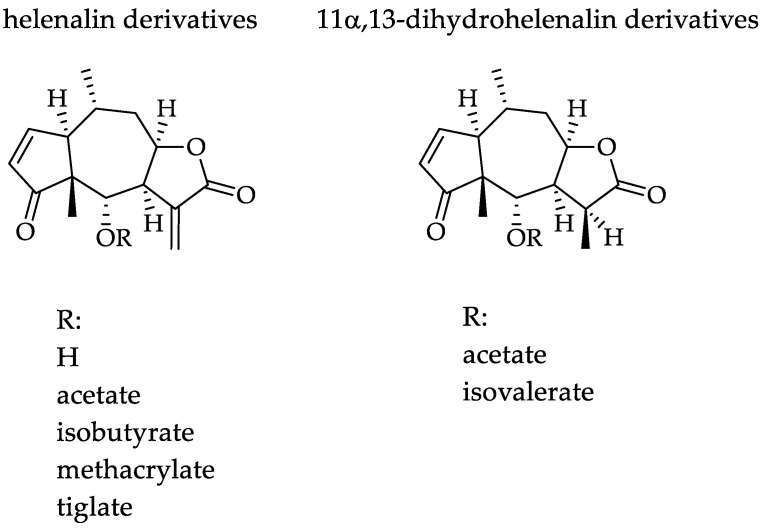
Chemical structures of the tested Arnica STLs.

**Figure 2 pharmaceuticals-15-00776-f002:**
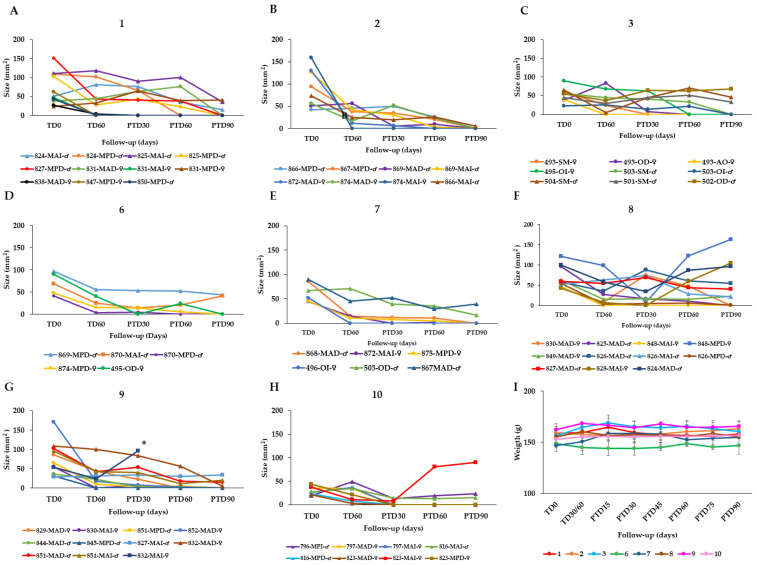
Effect of the topical AT treatment in golden hamsters with cutaneous leishmaniasis caused by *Leishmania braziliensis*. The figure shows the development of the lesion size in each hamster during the study in mm^2^ measured at various time points (graphics (**A**–**G**)). Disease progression was monitored before treatment administration (TD0), at the end of treatment (TD60) and on days 30, 60 and 90 post-treatment (PTD30, PTD60 and PTD90, respectively). (**A**): Group **1**; *L. braziliensis*, 19.2 μg total STL, 1× day, 60 days; (**B**): Group **2**; *L. braziliensis*, 19.2 μg total STL, 2× day, 60 days; (**C**): Group **3**; *L. braziliensis*, 38.4 μg total STL, 2× day, 60 days; (**D**): Group Group **6**; hamsters infected with *L. braziliensis* treated with 40 mg DOC Arnica 21,5% Creme (DAC; 4.0 μg total STL), 2× day, 60 days; (**E**): Group **7**; hamsters infected with *L. braziliensis* treated with 40 mg Arnica Salbe S Kneipp (ASK; 4.0 μg total STL), 2× day, 60 days; (**F**): Group **8**; hamsters infected with *L. braziliensis* treated with 40 mg STL enriched Arnica cream (EAC; 16.0 μg total STL), 2× day, 60 days; (**G**): Group **9**; hamsters infected with *L. braziliensis* treated with 40 mg STL enriched Arnica gel (EAG; 19.2 μg total STL), 2× day, 60 days; and (**H**): Group **10**; hamsters infected with *L. braziliensis* treated with meglumine antimoniate (MA), 200 μg/1× every 3 d, 30 d. Cure (●); improvement (▲); failure (■); endpoint (✶). The series in each panel correspond to the code of each hamster. (**I**): Development of the animals’ body weight of hamsters infected with *L. braziliensis* during the study. Changes in the body weight were monitored before treatment administration, at the end of treatment, and every two weeks until the study ended. TD0: Before treatment; TD30 for MA or TD60 for Arnica formulations: last day of treatment administration; PTD15: 15 days after the treatment ends; PTD30: 30 days after the treatment ends; PT45: 45 days after the treatment ends; PTD160: 60 days after the treatment ends; PTD75: 75 days after the treatment ends; and PTD90: 90 days after the treatment ends. Please note that detailed photographic documentation of the treatment progress in each group of hamsters is shown in Appendix A, Appendix A.

**Figure 3 pharmaceuticals-15-00776-f003:**
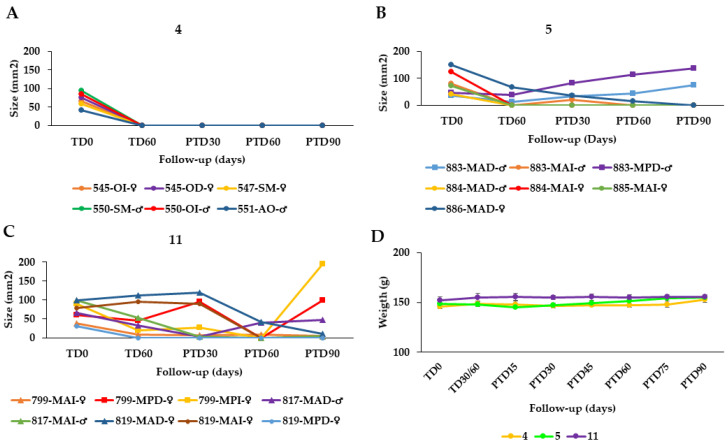
Effect of the topical AT treatment in golden hamsters with cutaneous leishmaniasis caused by *Leishmania tropica*. The figure shows the development of the lesion size in each hamster during the study in mm^2^ measured at various time points (graphics (**A**,**B**)). Disease progression was monitored before treatment administration (TD0), at the end of treatment (TD60) and on days 30, 60 and 90 post-treatment (PTD30, PTD60 and PTD90, respectively). (**A**): Group Group **4**; *L. tropica*, 19.2 μg total STL, 2x day, 60 days; (**B**): Group **5**; *L. tropica*, 38.4 μg total STL, 2× day, 60 days; (**C**): Group **11**; hamsters infected with *L. tropica* treated with meglumine antimoniate (MA), 200 μg/1× every 3 d, 30 d. Cure (●); improvement (▲); and failure (■). The series in each panel correspond to the code of each hamster. (**D**): Development of the animals’ body weight of hamsters infected with *L. tropica* during the study. Changes in the body weight were monitored before treatment administration, at the end of treatment, and every two weeks until the study ended. TD0: Before treatment; TD30 for MA or TD60 for Arnica formulations: last day of treatment administration; PTD15: 15 days after the treatment ends; PTD30: 30 days after the treatment ends; PT45: 45 days after the treatment ends; PTD160: 60 days after the treatment ends; PTD75: 75 days after the treatment ends; and PTD90: 90 days after the treatment ends. Please note that detailed photographic documentation of the treatment progress in each group of hamsters is shown in Appendix A, Appendix A.

**Figure 4 pharmaceuticals-15-00776-f004:**
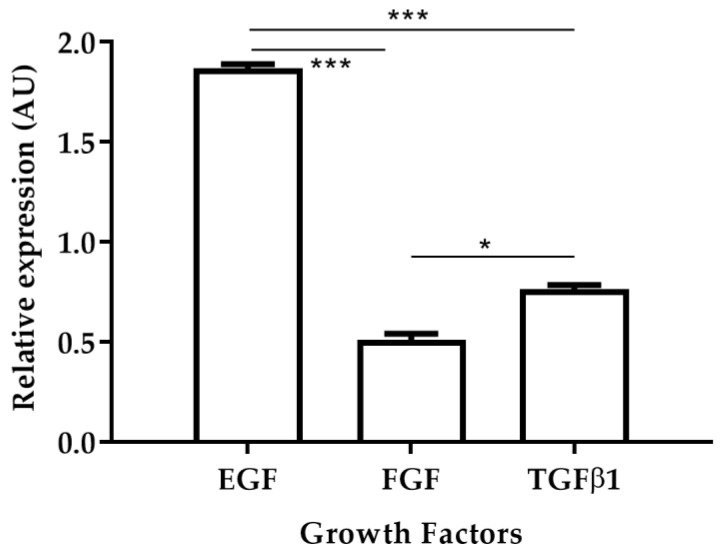
Relative expression of growth factors EGF, FGF and TGFβ1 after treatment with Arnica tincture. The figure shows the induction of growth factors calculated by the ΔΔCT methods between TD0 and PTD90 in cicatrized skins (*n* = 40). Bars represent the mean value ± SD. * *p* < 0.05; *** *p* < 0.001.

**Figure 5 pharmaceuticals-15-00776-f005:**
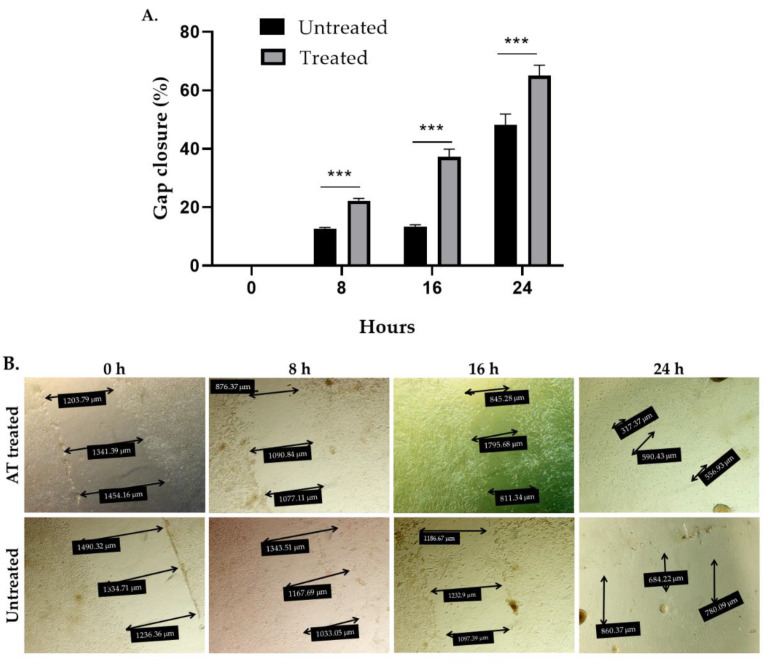
In vitro wound healing effect of Arnica tincture. The graph in (**A**) represents the mean ± standard deviation (*n* = 4) in the percentage of gap closure. *** Treatment with AT shows a statistically significant (*p* < 0. 001) increase in the percentage of gap closure in comparison to the untreated cells after 8 h, 16 h and 24 h of incubation vs. untreated cells. (**B**): Photographic documentation of a representative experiment showing the migration of cells into the gap area under exposure to Arnica tincture. The double arrows indicate cell free areas. Their diameters are reported in the black boxes.

**Table 1 pharmaceuticals-15-00776-t001:** Cytotoxic activity of Arnica tincture and isolated STLs. Data represent the mean lethal concentration (LC_50_) in μg/mL for each substance ± SD (*n* = 3 assays with 3 replicates in each case). U937: human tissue macrophages; Detroit 551: human skin fibroblasts; Hep G2: human liver cells.

Product	U937 LC_50_ (µg/mL)	Detroit 551 LC_50_ (µg/mL)	Hep G2 LC_50_ (µg/mL)
Tincture (total STLs)	>96	300 ± 4	380 ± 1
helenalin	5.75 ± 0.20	218 ± 44	19.1 ± 2.9
helenalin acetate	5.66 ± 0.08	59.8 ± 3.6	0.300 ± 0.080
helenalin isobutyrate	5.65 ± 0.01	69.5 ± 6.3	0.200 ± 0.010
helenalin methacrylate	5.68 ± 0.66	57.8 ± 9.2	1.20 ± 0.20
helenalin tiglate	5.80 ± 0.10	234 ± 44	52.2 ± 0.1
11α, 13-dihydrohelenalin acetate	8.23 ± 0.40	206 ± 35	41.0 ± 7.1
11α, 13-dihydrohelenalin isovalerate	10.1 ± 1.3	0.800 ± 0.020	0.600 ± 0.090
meglumine antimoniate	416 ± 23	492 ± 32	226 ± 29
doxorubicin	0.800 ± 0.100	2.00 ± 0.00	5.42 ± 0.10

**Table 2 pharmaceuticals-15-00776-t002:** Antileishmanial activity of Arnica tincture and isolated STLs. Data are expressed in µg/mL and represent means ± SD (*n* = 2 independent determinations with 3 replicates in each case). SI: Selectivity index = LC_50_ (cytotoxicity in macrophages)/EC_50_ (antileishmanial activity).

Product	*L. braziliensis*		*L. tropica*	
	EC_50_ (µg/mL)	SI	EC_50_ (µg/mL)	SI
Tincture (total STLs)	4.85 ± 0.19	>19.8	5.94 ± 1.17	>16.2
helenalin	2.02 ± 0.07	2.8	3.76 ± 0.78	1.5
helenalin acetate	1.86 ± 0.01	3.0	5.33 ± 0.17	1.1
helenalin isobutyrate	1.85 ± 0.03	3.1	2.20 ± 0.29	2.6
helenalin methacrylate	2.16 ± 0.13	2.6	2.40 ± 0.07	2.4
helenalin tiglate	2.14 ± 0.06	2.7	2.32 ± 0.14	2.5
11α, 13-dihydrohelenalin acetate	2.44 ± 0.10	3.4	3.50 ± 0.09	2.4
11α, 13-dihydrohelenalin isovalerate	3.04 ± 0.16	3.3	3.86 ± 0.29	2.6
meglumine antimoniate	13.7 ± 0.9	30.4	50.2 ± 1.3	8.3

**Table 3 pharmaceuticals-15-00776-t003:** Clinical outcome after treatment with Arnica tincture and semisolid Arnica formulations. Data represent the proportion (n animals/group size m) and percentage of hamsters for each outcome and each treatment. Cure: 100% re-epithelialization of the lesion. Improvement: any percentage reduction in lesion area. Failure: any increase in lesion size compared to baseline size. Relapse: reactivation of the lesion after initial cure during the study.

Formulation	*Leishmania* Species	Group	Dose	Cure n/m (%)	Improvement n/m (%)	Failure n/m (%)
Tincture	*L. braziliensis*	**1**	40 µL AT = 19.2 μg STL/1× d/60 d	8/11 (72.2)	2/11 (18.2)	1/11 (9.1)
		**2**	40 µL AT = 19.2 μg STL/2× d/60 d	7/8 (87.5)	1/8 (12.5)	0 (0)
		**3**	80 µL AT = 38.4 μg STL/2× d/60 d	6/9 (66.7)	2/9 (22.2)	1/9 (11.1)
	*L. tropica*	**4**	40 µL AT = 19.2 μg STL/2× d/60 d	6/6 (100)	0 (0)	0 (0)
		**5**	80 µL AT = 38.4 μg STL/2× d/60 d	5/7 (66.7)	0 (0)	2/7 (11.1)
DOC Arnica 21,5% Creme (DAC)	*L. braziliensis*	**6**	40 mg = 4.0 μg STL/2× d/60 d	4/11 (36.3)	2/11 (18.2)	5/11 (45.4)
Arnica Salbe S (Kneipp) (ASK)		**7**	40 mg = 4.0 μg STL/2× d/60 d	6/11 (54.5)	2/11 (18.2)	3/11 (27.3)
STL enriched Arnica cream (EAC)		**8**	40 mg = 16.0 μg STL/2× d/60 d	3/5 (60)	1 (20)	1 (20)
STL enriched Arnica Gel (EAG)		**9**	40 mg = 19.2 μg STL/2× d/60 d	4/6 (66.7)	2/6 (33.3)	0 (0)
Positive controls: Meglumine antimoniate (MA; intralesional injection)	*L. braziliensis*	**10**	200 μg/every 3 d/30 d	5/8 (62.5)	1/8 (12.5)	2/8 (25)
	*L. tropica*	**11**	200 μg/every 3 d/30 d	3/8 (37.5)	2/8 (25)	3/8 (37.5)

**Table 4 pharmaceuticals-15-00776-t004:** Changes of the serum levels of alanine transaminase (ALT), blood urea nitrogen (BUN) and creatinine in golden hamsters before and after treatment with Arnica tincture and other topical formulations as well as meglumine antimoniate. Data represent means ± SD for each group of hamsters. R.V.: reference values.

Group	ALT (U/L) (R.V. 22–128)		CREATININE (mg/dL) (R.V. 0.4–1.0)		BUN (mg/dL) (R.V. 12–26)	
	TD0	TD8	TD0	TD8	TD0	TD8
**1**	62.7 ± 4.6	78.3 ± 11.8	0.4 ± 0.02	0.5 ± 0.05	19.0 ± 2.7	18.7 ± 0.6
**2**	57.0 ± 8.7	80.3 ± 13.3	0.4 ± 0.01	0.5 ± 0.02	24.8 ± 3.6	22.7 ± 0.6
**3**	45.3 ± 5.0	61.2 ± 1.8	0.5 ± 0.02	0.5 ± 0.03	14.2 ± 2.4	23.6 ± 2.3
**4**	58.1 ± 2.6	75.4 ± 4.8	0.4 ± 0.04	0.4 ± 0.03	15.3 ± 1.3	21.3 ± 4.8
**5**	47.4 ± 1.2	68.7 ± 4.4	0.6 ± 0.01	0.3 ± 0.02	21.3 ± 3.3	17.5 ± 6.8
**6**	57.3 ± 4.7	61.3 ± 4.2	0.7 ± 0.02	0.4 ± 0.01	16.2 ± 4.7	22.1 ± 3.3
**7**	67.2 ± 3.4	73.4 ± 5.6	0.5 ± 0.03	0.5 ± 0.03	15.7 ± 2.1	19.5 ± 1.7
**8**	60.5 ± 2.3	75.3 ± 5.6	0.4 ± 0.02	0.3 ± 0.02	24.2 ± 1.5	21.0 ± 2.8
**9**	64.9 ± 3.7	58.2 ± 2.9	0.6 ± 0.03	0.4 ± 0.03	23.3 ± 1.1	20.3 ± 2.7
**10**	43.3 ± 11.3	49.5 ± 10.9	0.5 ± 1.0	0.5 ± 0.05	21.1 ± 2.5	23.3 ± 2.3
**11**	48.5 ± 7.3	62.0 ± 14.5	0.5 ± 0.08	0.05 ± 0.07	20.3 ± 3.8	19.8 ± 3.1

## Data Availability

Data reported in this study are contained within the article. The underlying raw data are available on request from the corresponding author.

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
