# Peer review of "Therapeutic Efficacy of Arnica in Hamsters with Cutaneous Leishmaniasis Caused by Leishmania braziliensis and L. tropica"

_pharmaceuticals, 2022, doi:10.3390/ph15070776_

Round 1

Reviewer 1 Report

In manuscript pharmaceuticals-1751341, the authors have carried out in-vitro and in-vivo investigations of the efficacy of Arnica montana extract formulations against leishmaniasis. The Arnica treatment protocols offer promising complementary or alternative treatment for cutaneous leishmaniasis and is worth pursuing further.

Would a formulation with a better skin-penetrating ingredient (e.g., DMSO) provide better results? Have the authors considered other formulations of Arnica?

Some minor editorial corrections:

Line 19: Asteraceae should not be italicized.

Lines 25-26: “The in-vivo therapeutic effect…” [use a hyphen for double modifiers]

Lines 28-29: “…a very high in-vitro activity…” [insert hyphen]

Line 67: Use either “Not only pentavalent antimonials but also other…” OR “Pentavalent antimonials in addition to other…”

Lines 94-95: “…the in-vitro antileishmanial activity…” [insert hyphen]

Lines 100-101: “…for in-vivo efficacy…” [insert hyphen]

Lines 114-116: Do not capitalize chemical compound names.

Line 157: re-epithelialization [insert hyphen]

Author Response

Reviewer #1:

Would a formulation with a better skin-penetrating ingredient (e.g., DMSO) provide better results? Have the authors considered other formulations of Arnica?

It is certainly possible that such additives may lead to higher penetration rates. However, we chose to test the traditional herbal preparation which is already approved and in use for centuries so that its safety in humans has already been proven. The very favorable in vivo results, moreover, prove that our “repositioning” approach is very promising for the treatment of CL. It is also interesting to note that we did test other formulations (creams, gel), which were less potent than the traditional tincture.

Some minor editorial corrections:

Line 19: Asteraceae should not be italicized.

This was corrected.

Lines 25-26: “The in-vivo therapeutic effect…” [use a hyphen for double modifiers]

It appears that the journal does not write in-vivo/in-vitro but both expressions without hyphens. (Checked in several recent articles).

Lines 28-29: “…a very high in-vitro activity…” [insert hyphen]

It appears that the journal does not write in-vivo/in-vitro but both expressions without hyphens. (Checked in several recent articles).

Line 67: Use either “Not only pentavalent antimonials but also other…” OR “Pentavalent antimonials in addition to other…”

This was corrected.

Lines 94-95: “…the in-vitro antileishmanial activity…” [insert hyphen]

It appears that the journal does not write in-vivo/in-vitro but both expressions without hyphens. (Checked in several recent articles).

Lines 100-101: “…for in-vivo efficacy…” [insert hyphen]

It appears that the journal does not write in-vivo/in-vitro but both expressions without hyphens. (Checked in several recent articles).

Lines 114-116: Do not capitalize chemical compound names.

This was corrected everywhere (also in Figure 1 and in the tables for consistency).

Line 157: re-epithelialization [insert hyphen]

This was corrected.

We thank the reviewer for the time and efforts to help us improve our manuscript!

Reviewer 2 Report

The present study attempted to investigate the anti-Leishmania effect of Arnica (an ethanolic extract and sesquiterpene lactones). This is an interesting work and specific comments are as follows.

 HPLC data or other chemical analyses of the ethanolic extract and sesquiterpene lactones should be added to the manuscript.

 Details for experiments should be described in the tables (time of treatment, cell number etc., Tabs 1 and 2) (parasite burden, Tab 2).

 Please, include some morphological aspects of the treated and non treated cells (U937, Hep G2, and human skin fibroblasts), such as their shape and appearance, evaluated by optical microscopy.

 Please change Fig.2 and add the evolution of cutaneous lesion in non treated or mock-treated animals, and consider adding two figures (Fig. 2 for L. tropica infection and Fig. 3 for  L. braziliensis infection).

 .Did the authors treat L. tropica infected hamsters with STLs?

 The authors could present results related to serum enzyme and creatinine levels and

(no)  macroscopic alterations in organs or tissues of hamsters (lines 286-289).

 Histopathological analyses could be performed to evaluate cell and tissue alterations and semi-quantitative assessment of amastigotes.

Author Response

Reviewer #2

Comments and Suggestions for Authors

The present study attempted to investigate the anti-Leishmania effect of Arnica (an ethanolic extract and sesquiterpene lactones). This is an interesting work and specific comments are as follows.

HPLC data or other chemical analyses of the ethanolic extract and sesquiterpene lactones should be added to the manuscript.

An UHPLC- QqTOF MS chromatogram of the investigated Arnica tincture and 1H NMR spectra of the individual tested STLs have been added as Figures S12-S19 in the supplementary materials.

- Details for experiments should be described in the tables (time of treatment, cell number etc., Tabs 1 and 2) (parasite burden, Tab 2).

The requested information was added:

Table 1.: Cytotoxic activity of Arnica tincture and isolated STLs. Data represent the mean lethal concentration (LC50) in μg/mL for each substance ± SD (n = 3 assays with 3 replicates in each case). U937: human tissue macrophages; Detroit 551: human skin fibroblasts; Hep G2: human liver cells. 10,000 cells/well were exposed to six 1:2 dilutions of arnica tincture and STLs for 72 hours under incubation at 37 °C, 5% CO2. Cytotoxicity was assessed using MTT method.

Table 2.: Antileishmanial activity of Arnica tincture and isolated STLs. Data are expressed in µg/mL and represent means ± SD (n=2 independent determinations with 3 replicates in each case). SI: Selectivity index =LC50 (cytotoxicity in macrophages)/EC50 (antileishmanial activity).  First, U937 cells (3x105) were first infected with L. braziliensis or L. tropica promastigotes (45x105) for 24 hours by incubation at 34 °C, 5% CO2. Secondly, infected cells were exposed to six 1:2 dilutions of arnica tincture and STLs for 72 hours under incubation at 37 °C, 5% CO2. Parasite load was assessed by flow cytometry by counting green fluorescent parasites. 

- Please, include some morphological aspects of the treated and non treated cells (U937, Hep G2, and human skin fibroblasts), such as their shape and appearance, evaluated by optical microscopy.

The U937 cells have the morphology of monocytes with a round appearance and grow as suspension cells. Hep G2 cells are epithelial cells of varying size and shape, but predominantly polygonal in shape; Detroit 551 cells are fibroblast-like cells, variable in shape and size; they may be spindle-shaped or stellate with cytoplasmic extensions that may be relatively short and broad, or long, thin and highly branched. Both Hep G2 and Detroit 551 cells grow as adherent cells forming a monolayer. Once the cell cultures are exposed to cytotoxic concentrations, the appearance of the U937 cells becomes irregular, with stellate and rough edges, and the Hep G2 and Detroit 551 cells detach from the monolayer and lose their extensions.

This information has been added to section 3.3. in the experimental section.

Please change Fig.2 and add the evolution of cutaneous lesion in non treated or mock-treated animals, and consider adding two figures (Fig. 2 for L. tropica infection and Fig. 3 for  L. braziliensis infection).

The Ethics Committee for animal experimentation does not allow the inclusion of a group of infected animals to which no treatment is given. In the specific case of leishmaniasis, hamsters that develop a lesion larger than 4 mm after experimental infection are entered into the study. This ensures that all animals have the disease established and are subjected to different treatments. 

Although it is believed that CL can resolve without treatment, our experience of more than 30 years working in the field of human leishmaniasis has shown us that lesions which occasionally manage to resolve are those lesions that are just beginning to develop. That is when the infection is establishing itself.  On the contrary, cutaneous lesions, especially ulcers, resolve only after having received empirical or conventional treatment.

As requested by the reviewer, the previous Figure 2 was divided into  two figures: Figure 2 with results of L. braziliensis and  Figure 3 with results of L. tropica. The Figure numbers were revised in the text accordingly.

Did the authors treat L. tropica infected hamsters with STLs?

If the reviewer means isolated STLs, the answer is no. Only Arnica tincture and the mentioned semi-solid products were tested in vivo in this study as mentioned and justified in the text. However, in case of each tested product, the STL concentration was determined (and is mentioned in the text).

The authors could present results related to serum enzyme and creatinine levels and (no)  macroscopic alterations in organs or tissues of hamsters (lines 286-289).

Table 4 was added.

Table 4.: Changes of the serum levels of alanine transaminase (ALT), blood urea nitrogen (BUN) and creatine in golden hamsters before and after treatment with Arnica tincture and other topical formulations and meglumine antimoniate. Data represent means ± SD for each group of hamsters. R. V: reference values.

Group

ALT (U/L)

(R.V 22 – 128)

CREATININE (mg/dL)

(R.V 0.4 – 1.0)

BUN (mg/dL)

(R.V 12 – 26)

TD0

TD8

TD0

TD8

TD0

TD8

1

62.7 ± 4.6

78.3 ± 11.8

0.4 ± 0.02

0.5 ± 0.05

19.0 ± 2.7

18.7 ± 0.6

2

57.0 ± 8.7

80.3 ± 13.3

0.4 ± 0.01

0.5 ± 0.02

24.8 ± 3.6

22.7 ± 0.6

3

45.3 ±

61.2 ±

0.5 ± 0.02

0.5 ± 0.03

14.2 ± 2.4

23.6 ± 2.3

4

58.1 ±

75.4 ±

0.4 ± 0.04

0.4 ± 0.03

15.3 ± 1.3

21.3 ± 4.8

5

47.4 ±

68.7 ±

0.6 ± 0.01

0.3 ± 0.02

21.3 ± 3.3

17.5 ± 6.8

6

57.3 ±

61.3 ±

0.7 ± 0.02

0.4 ± 0.01

16.2 ± 4.7

22.1 ± 3.3

7

67.2 ±

73.4 ±

0.5 ± 0.03

0.5 ± 0.03

15.7 ± 2.1

19.5 ± 1.7

8

60.5 ±

75.3 ±

0.4 ± 0.02

0.3 ± 0.02

24.2 ± 1.5

21.0 ± 2.8

9

64.9 ±

58.2 ±

0.6 ± 0.03

0.4 ± 0.03

23.3 ± 1.1

20.3 ± 2.7

10

43.3 ± 11.3

49.5 ± 10.9

0.5 ± 1.0

0.5 ± 0.05

21.1 ± 2.5

23.3 ± 2.3

11

48.5 ± 7.3

62.0 ± 14.5

0.5 ± 0.08

0.05 ± 0.07

20.3 ± 3.8

19.8 ± 3.1

Histopathological analyses could be performed to evaluate cell and tissue alterations and semi-quantitative assessment of amastigotes.

Histopathological analyses were only performed to evaluate the safety profile (cell and tissue alterations attributed to the AT products). The presence of amastigotes was tested by RT-PCR.

We thank the reviewer for the time and efforts to help us improve our manuscript!

Reviewer 3 Report

The manuscript entitled “Therapeutic efficacy of Arnica in hamsters with cutaneous leishmaniasis caused by Leishmania braziliensis and L. tropica” deals with a very interesting topic, that of the use of Arnica in hamsters with cutaneous Leishmaniasis. This topic fits within framework in which current drugs used “pentavalent antimonial”  are the leading treatment for cutaneous leishmaniasis despite the hepatic, renal, and cardiac toxicity. In this work, the author studied the in vitro cytotoxicity and antileishmanial activity of AT and STLs against both L. braziliensis and L. tropica and then they evaluated the in vivo therapeutic effect of AT in hamsters with cutaneous Leishmaniasis (CL) caused by experimental infection with L. braziliensis and L. tropica. The article is well written and well structured, the experiments are well conducted and create interest for the reader. However, there are some observations that I would like the authors to make explicit:

-          in the whole text the words “in vivo” and “in vitro” should be put in italics: lines 23,24,26,94,97,100,388,389

-          In section 2.1. from line 117 to line 125, verb tenses should be in the “past tense”

-          the images show badly ( even those of the supplementary material).

Author Response

Reviewer #3:

The manuscript entitled “Therapeutic efficacy of Arnica in hamsters with cutaneous leishmaniasis caused by Leishmania braziliensis and L. tropica” deals with a very interesting topic, that of the use of Arnica in hamsters with cutaneous Leishmaniasis. This topic fits within framework in which current drugs used “pentavalent antimonial”  are the leading treatment for cutaneous leishmaniasis despite the hepatic, renal, and cardiac toxicity. In this work, the author studied the in vitro cytotoxicity and antileishmanial activity of AT and STLs against both L. braziliensis and L. tropica and then they evaluated the in vivo therapeutic effect of AT in hamsters with cutaneous Leishmaniasis (CL) caused by experimental infection with L. braziliensis and L. tropica. The article is well written and well structured, the experiments are well conducted and create interest for the reader. However, there are some observations that I would like the authors to make explicit:

-          in the whole text the words “in vivo” and “in vitro” should be put in italics: lines 23,24,26,94,97,100,388,389

It appears that the journal does not write in vivo/in vitro in italics (Checked in several recent articles).

-          In section 2.1. from line 117 to line 125, verb tenses should be in the “past tense”

Past tense was used where appropriate.

-          the images show badly ( even those of the supplementary material).

This may be related to the conversion into a pdf. The images are in the best possible quality in the submitted word document and we also submitted them separately as *.eps files so that they can be of better quality in the final publication.

We thank the reviewer for the time and efforts to help us improve our manuscript!

Round 2

Reviewer 2 Report

The authors did the appropiate changes.